# Biological Insights on the Invasive Fig Pest *Aclees taiwanensis* Kȏno, 1933 (Coleoptera: *Curculionidae*)

**DOI:** 10.3390/insects14030223

**Published:** 2023-02-23

**Authors:** Camilla Tani, Barbara Conti, Stefano Bedini

**Affiliations:** Department of Agriculture, Food and Environment University of Pisa, 80 Via del Borghetto, 56124 Pisa, Italy

**Keywords:** fig weevil, artificial diet, biological parameters, invasive pest

## Abstract

**Simple Summary:**

*Ficus carica* L. (Moraceae) is a particularly ancient cultivated fruit plant, characteristic of the Mediterranean diet and landscape. The fig weevil *Aclees taiwanensis* (Coleoptera: *Curculionidae*) is a new phytophagous that has been threatening Mediterranean figs in recent years. *A. taiwanensis* is a Coleoptera native to Asia, firstly reported in France in 1997 as *A. cribratus*, and in Italy in 2005 as *A.* sp. cf. *foveatus*. The major damage is caused by the larvae which dig tunnels in the stems and roots of the fig tree, compromising the phloem flow of the plant. Unfortunately, no method for the control of the fig weevil has proven effective. In fact, information on the insect’s biology and behavior is limited to that obtained from adult specimens collected in the field, and little information is available on larval stages. The purpose of this study, therefore, was to fill in the gaps on the biology of *A. taiwanensis* by developing a breeding protocol that is easy and inexpensive.

**Abstract:**

The fig weevil *Aclees taiwanensis* Kȏno, 1933 (Coleoptera: *Curculionidae*) is an invasive fig tree pest recently introduced in southern Europe. Reported for the first time in France in 1997 as *A. cribratus*, and then in Italy in 2005 as *A.* sp. cf. *foveatus*, *A. taiwanensis* is currently threatening fig nurseries, orchards, and wild plants. To date, no control methods have proven to be effective against *A. taiwanensis*. Although some attempts have been made to describe the insect’s biology and behavior, such information is limited to that obtained from adult samples collected in the field. In particular, because of their xylophagous behavior, scarce information is available on the larval stages of the species. The aim of this study, therefore, was to fill these information gaps on the insect biology and behavior by setting up a laboratory protocol suitable for the rearing of *A. taiwanensis*. Using the developed rearing protocol, we assessed the main fitness parameters of the species including oviposition rate, egg hatchability, embryonic, larval and pupal duration and development, immature survival, pupation behavior, pupal weight, emergence, sex ratio and adult morphological parameters. The proposed rearing procedure allowed us to obtain new information on the main features of the insect’s biology that may be useful for setting up strategies for its control.

## 1. Introduction

The spread of invasive species from tropical to temperate areas is an emerging issue due to globalization and climate change. In the last 30 years, more than 12,000 ‘alien’ species have been introduced to Europe, with more than 3000 in Italy alone, and with a dizzying increase of 96% in the last 10 years. Among invertebrates, 90% is represented by insect pests causing huge economic losses that have been estimated as more than 76.0 billion US $ per year [1]. The fig weevil *Aclees taiwanensis* (Coleoptera: *Curculionidae*) [1,2] is an invasive pest harmful to the fig tree *Ficus carica* (Moraceae). The species, native to subtropical and tropical Asia, threatens southern European fig nurseries, orchards, and wild plants, and has also recently been reported in South Korea [3,4]. In Europe, *A. taiwanensis* was first recorded as *A. cribratus* in 1997 in France [5]. In 2005, *A. taiwanensis* was reported as *A.* sp. cf. *foveatus* in fig nurseries in Italy [6], and to date, it is spreading to various regions of Italy and France [7]. In the Mediterranean area, *A. taiwanensis* represents a major threat to the large germplasm variety of fig trees [7], one of the earliest cultivated fruit trees [8] that, along with the olive tree, are iconic of the Mediterranean landscape and agriculture. Moreover, figs are a typical component of the Mediterranean diet [9], considered one of the healthiest fruits associated with longevity [4] and used in traditional medicine for their laxative, cardiovascular, respiratory, antispasmodic, and anti-inflammatory properties [5].

The attack of *A. taiwanensis* on fig orchards is critical and destructive. Not only do adults damage the fig trees by feeding on buds, leaves and immature fruits, but the xylophagous larvae enter the tree trunks and roots by digging tunnels that destroy the plant xylem and phloem [10], causing plant death. Unfortunately, the difficulty in early attack detection, together with the difficulty to reach the larvae inside the wood, represent the main problems in the control of *A. taiwanensis*, and to date, no control methods have proven to be effective against *A. taiwanensis*. Furthermore, the lack of specific EU regulation prevents effective actions to counter the *A. taiwanensis* spread among countries via plant trading [4], which may facilitate a rapid diffusion of this pest all over the Mediterranean area where fig trees are cultivated. Thus, effective strategies to detect and control the black weevil are urgently needed.

Previous attempts to control the pest by insecticides [6,10] and entomopathogenic fungi [11] were unsuccessful, mainly because the larvae, well protected inside the trunk, are not vulnerable to the direct delivery of insecticides. In this scenario, a detailed knowledge of the insect biology, behavior, and nutritional requirements is fundamental for the development of new methods to tackle *A. taiwanensis*. However, despite the serious threat to fig nurseries and orchards posed by *A. taiwanensis*, very little data is available on the insect’s biology, distribution and actual impact on fig production [7], and more than half of the data available in the literature are more than 10 years old. In particular, because of their xylophagous behavior, very scarce information is available on the larval stages of the species. In this context, the aim of this study was to set up a rearing technique for *A. taiwanensis* based on the use of an artificial diet for the larvae and to contribute to filling the information gap on the insect’s biology.

## 2. Materials and Methods

### 2.1. Insect Rearing

*Aclees taiwanensis* first instar larvae were obtained from wild specimens (three couples) collected in the open field in May 2019. Specimens were collected in fig orchards located in Carmignano (Prato, Italy, 43°48′36.97″ N; 11°00′53.78″ E), sexed according to Farina et al. [7] and maintained under laboratory conditions (30 ± 1 °C, 60 ± 5% relative humidity, and 16:8 L:D photoperiod). Each couple was kept in a separate cylindrical Plexiglas cage (25 cm diameter, 40 cm high, and top opening covered by a net). A pot (13 × 10 × 10 cm^3^) filled with topsoil was placed in each cage and covered with leaves as a shelter. A fig twig (10 cm length, diameter 1 cm) was inserted in each pot as an oviposition site. Each cage was provided with apple slices and fig leaves. Water was supplied by means of a roll of gauze dipped into a falcon test tube with a pierced stopper. The fig sprigs were checked weekly to verify oviposition, and those with eggs laid were changed with fresh sprigs. To obtain the larvae, the eggs were gently collected with a fine moistened brush, rinsed to remove organic residues, and placed in a Petri dish (100 mm in diameter) containing a Whatman paper (90 mm in diameter) moistened with tap water. Larvae were reared on artificial media as follows: fresh fig branches collected in Usigliano di Lari (Pisa, Italy, 43°34′8″ N; 10°35′38″ E) were ground by an electric blender. The coarse powder was frozen at −20 °C for 24 h and ground again for 5 min. The larger wood fragments were removed from the sawdust by sieving (Sieve Premier Housewares 0806512) and the fig sawdust was finally stored at −20 °C until use. For the diet preparation, 32 g of agar were dissolved in 500 mL of deionized water in a microwave oven (6 min, 750 W). After agar solubilization, the volume was adjusted with deionized water to 800 mL and allowed to cool down to 50 °C. Then, the fig sawdust (200 g) and the other ingredients: α-cellulose, 100 g; yeast powder, 28 g; sucrose, 40 g; wheat germ, 40 g; Wesson’s salt, 12 g; ascorbic acid, 4 g; cholesterol, 4.8 g; nipagin, 2 g; vitamins, 8 mL (vitamin supplement containing 100 g of product: thiamine hydrochloride, 10 mg, riboflavin, 5 mg, pyridoxine hydrochloride, 5 mg, nicotinamide, 100 mg, calcium pantothenate, 5 mg, p-aminobenzoic acid, 2.5 mg, choline chloride, 125 mg, inositol, 5 mg), were added and the solution was adjusted with deionized water to 1000 mL and gently stirred for about 3 min. Media and supplements were supplied by Sigma Italia (Milano, Italy). Finally, the agarized medium was distributed in 350 mL sterile plastic jars. An amount of 50 mL (for first- and second-instar larvae) or 100 mL of diet (for older larvae) was poured into the jars and allowed to solidify at room temperature. After cooling, the jars were sealed with Parafilm M (Heathrow Scientific, Vernon Hills, IL, USA) and stored at 4 °C. Two newly emerged larvae were placed with a fine brush in each jar containing 50 mL of the diet previously prepared. To facilitate their entry into the medium, some holes were opened by a bradawl on the medium surface. Each jar was closed by a sterile gauze to protect the surface and to allow air exchange. The jars were kept in the dark at 30 ± 1 °C and 60 ± 5% RH and inspected daily for molting by checking for the presence of larvae cephalic capsules. Larvae of the last instar were left undisturbed to allow the pupation inside a pupal chamber to build into the agarized diet. After adult emergence, *A. taiwanensis* males and females were kept separately in cylindrical Plexiglas cages (25 cm diameter, 40 cm high and upper opening covered with gauze), under laboratory conditions (30 ± 1 °C, RH 60 ± 5%, and photoperiod 16:8 L:D) with water and food as reported above (see egg production). Water and fresh food were changed three times a week.

### 2.2. Larval and Pupal Biological Parameters Assessment

The number of larval stages, as well as their duration (pupal stage included), were assessed on 30 newly emerged larvae that reached the pupal instar. Starting from the third larval stage, every 10 days, the larvae were transferred to jars containing 100 mL of new diet. The larvae were weighed using a precision balance and were measured in length. The pupal stage is very delicate and does not allow for manipulation. Any attempt to measure length, weight or to observe in depth the pupae had a very negative impact on their survival. The only observations allowed concerned a general observation without manipulation of the external morphology from which it was possible to establish no sexual dimorphism.

### 2.3. Adult Morphological Parameters and Sex Ratio

After emergence, the adults were weighed using a precision balance (Kern ABS, Kern & Sohn GmbH, Balingen, Germany) and were measured in length. To evaluate the performance of the rearing method, the weight and length data of 10 males and 10 females, randomly chosen, were compared to the data of wild specimens (10 males and 10 females) collected in open field (see above). The experiment was repeated three times (in July, August and September 2021) for a total of 30 males and 30 females. To calculate the species sex ratio, all the adults obtained were sexed using the method described by Farina et al. [7].

### 2.4. Fecundity

To determine *A. taiwanensis*’ fecundity, twelve newly emerged virgin females were allowed to mate with twelve sexually mature males (for a total of twelve couples). Each couple was maintained under the same laboratory conditions in cages provided with water, food, and fig twigs for oviposition. A new fig twig was provided to each insect couple weekly. Oviposition was checked daily after the mating and three times a week for 12 months after the first laid egg. The laid eggs, as in the above reported protocol (see paragraph 2), were collected, counted, and placed in a Petri dish (100 mm in diameter) containing a Whatman paper (90 mm in diameter) moistened with deionized water. The Petri dishes, kept under laboratory conditions, were checked daily for newly emerged larvae and the total number of hatched eggs was recorded. 

### 2.5. Statistical Analysis

Differences in biological parameters between laboratory-reared (LR) and wild individuals, reared in the period November 2020–December 2021, were assessed by two-tailed Student’s *t*-test. Statistics were performed with GraphPad Prism 8 software (GraphPad Software, Inc.; San Diego, CA, USA).

## 3. Results

### 3.1. Insect Rearing

The formulated artificial diet resulted suitable to sustain the developing cycle of *A. taiwanensis* from eggs to adults. Of the 800 larvae fed on the artificial diet, approximately 39.3% pupated, and from these pupae, 49.5% adults developed (Table 1).

We modulated the preservatives to inhibit molds and bacteria growth, while allowing the survival of the larvae. In fact, according to our observations, no visible molds or change in color were observable in the diet until ten days under laboratory conditions and, at 4 °C it remained well-preserved for at least a month.

### 3.2. Larval and Pupal Biological Parameters Assessment

Using the artificial diet formulated in this experiment, *A. taiwanensis* pupation occurred after five larval instars (Table 2). On average, *A. taiwanensis* egg hatching occurred after approximately nine days, and larvae completed their development and pupated in approximately 41 days (Table 2). Adults emerged approximately 12 days after pupation (Table 2). Overall, in the controlled environmental conditions of this experiment, *A. taiwanensis* adults emerged approximately 62 days after oviposition.

The duration of the larval stages ranged from 3.3 to 22.7 days for L1 and L5, respectively (Table 3).

Larval weight ranged from 0.01 to 0.40 g for L2 and L5, respectively, while larval length ranged from 0.30 to 1.83 cm for L1 and L5, respectively. No significant difference was observed between the length of larvae reared on an agarized diet and on fig twigs (Table 4).

Before molting, fifth-instar larvae stopped digging galleries in the substrate and built a pupation chamber. After molting, *A. taiwanensis* pupae were whitish and became caramel-colored by the third day (Figure 1). Approximately five days before the adult emergence, the eyes of the pupae turned red and then, approximately two days before emergence, the eyes of the pupae turned black. In a few cases, the fifth-instar larvae were not able to build a proper pupation chamber. This resulted in abnormal development of the adult’s wings and premature death.

### 3.3. Adult Morphological Parameters and Sex Ratio

Adults, after emergence, remained inside the pupal chamber for approximately two days. As expected, the cuticle of the adults changed in color after the emergence. The newly emerged adults were whitish and then turned black through a progressive series of caramel-brown shades. The complete color change of adults, from white to black, occurred in approximately eight days.

Among the adults that emerged from the reared pupae, females represented 61% (sex ratio = 1.5:1). On average, the weight of the females was 0.27 g ± 0.60, while the average weight of the males was 0.21 g ± 0.074. Though males were significantly lighter than females (*t* = 3.101, df = 58, *p* = 0.003), no significant differences in body length were found between the sexes (*t* = 1.817, df = 60, *p* = 0.074). No significant differences in the weight and length of females and males were found between wild specimens and adults reared on the agarized diet (AD) (*t* = 1.691, df = 60, *p* = 0.097) (Figure 2).

Interestingly, after the color change, we observed the appearance of spots of pinkish-beige waxy powder both in males and females. Such powder was distributed over the upper and lower side of the individuals, mostly concentrated at the terminal part of the elytra and under the head as well as on the legs (Figure 3).

According to our observations, the pinkish powder occurred after 15.96 ± 5.25 days from emergence. 

Overall, 39 out of 156 adults produced by the laboratory-rearing procedure during the one-year observation period died (25% of mortality), suggesting that the average life expectancy of *A. taiwanensis* can exceed one year.

### 3.4. Fecundity

The fecundity experiment showed that females were able to lay the first egg 71.00 ± 10.06 days after emergence from the pupal instar. During the year of observation, the highest egg production occurred in March with an average oviposition rate of 30.25 ± 11.12 eggs/female (Table 4). The monthly eggs laid by the 12 observed couples is reported in Figure 4.

## 4. Discussion

Climate change and globalization have considerably enhanced the frequency of events that introduce dangerous alien species from subtropical and tropical regions to temperate. In regards to insect pests, deep knowledge of a newly introduced species’ biology and behavior is paramount to preventing and reducing the ecological and economic impact of such events. In particular, artificial rearing systems that allow insect development along their entire life cycle are fundamental to provide an adequate number of specimens to be utilized in biological, behavioral and toxicological studies [12,13,14,15], and are essential not only for the description of the insect’s features, but also for the development of efficient control programs. Currently, due to the difficulties of rearing in laboratory conditions, the fig tree pest *A. taiwanensis’* biology is still poorly described. In addition, available information is limited to that obtained from adult samples collected in the field. In a previous work [7], we managed to rear *A. taiwanensis* by rearing the larvae in twigs of the host plant *F. carica*. However, the procedure was cumbersome and time consuming. In addition, the high larval mortality resulted in an adult production that was too scarce for any experimental purpose. In contrast, in this work, the high percentage of developed larvae (69.2%), as well as the lack of any significant differences in morphometric parameters between laboratory-reared and wild adults, indicate that the developed protocol by artificial diet for the larval stages is suitable for the rearing of *A. taiwanensis* in amounts sufficient for detailed studies.

In this study, the diet for the rearing of the *A. taiwanensis* larvae was developed in accordance with the formulation of artificial diets previously utilized for other Coleoptera species [13,14,16,17]. For the formulation in this experiment, we added fig wood sawdust to make the agarized substrate more similar to the natural environment of *A. taiwanensis* larvae. A similar strategy was previously adopted by Mizuno and Kajimura [18], by adding Douglas-fir sawdust to the formulation of semi-artificial diets for rearing the ambrosia beetle *Xyleborus pfeili*. Castrillo et al. [19] obtained brood production by the ambrosia beetle *Xylosandrus germanus* reared on an artificial diet based on American beech *Fagus americana*, black walnut *Juglans nigra*, European buckthorn *Rhamnus cathartica*, and red oak *Quercus rubra* sawdust. The use of an artificial media containing sawdust was also successful for rearing the ambrosia beetles *Xyleborus volvulus* and *Xyleborus bispinatus* and for studying their biology [20,21].

The number of pre-imaginal *A. taiwanensis* instar observed in this experiment is consistent with that observed for other Curculuionidae, such as *Sitona hispidulus*, *Hylobitelus xiaoi*, and *Pagiophloeus tsushimanus* [22,23,24], where pupation occurred after five larval instars. Similarly, the length of the laboratory-reared larvae was about the same of those reared in fig twigs (Table 2), suggesting that the artificial diet did not affect this aspect of larval development. 

In this experiment, pupation occurred after about 41 days while, the total time of development of larvae reared in fig twigs was approximately 77 days [7]. These differences were probably due to the different temperatures under which the insects were kept. In fact, Farina and co-authors [7] kept the insects at 25 ± 1 °C, while in the present work they were maintained at 30 ± 1 °C. Environmental conditions are known to also influence larval and embryonic duration [25,26,27,28,29]. However, the absence of any larvae handling in this experiment may have also positively affected the duration of the development. In contrast, in Farina et al. [7] the handling required to extract them from the fig twigs may have affected the duration of larval development. It is well known that manipulation can disturb the development of insects, and in general increase, as a reaction, the duration of the development time [30].

In our experiment, during the last instar, the larvae of *A. taiwanensis* almost stopped feeding and remained without moving for approximately four days inside a chamber within the diet. During this stage, the diet was not renewed and the larvae were left undisturbed, as suggested by El-Shafie et al. [30] for the red palm weevil, *Rhynchophorus ferrugineus*. 

In line with what was observed in other *Curculionidae*, no sexual dimorphism was evidenced at the pupal stage [31].

Interestingly, after emergence, we observed the appearance of pinkish-beige, waxy powder spots both in males and females. As in *A. taiwanensis,* specific colorations have been observed in other *Curculionidae* [32,33], although their function has not been clarified. In the case of *A. taiwanensis* it can be hypothesized that the waxy dust may have a mimetic role in the clayey and dry soils during the summer and on the bleached fig bark, or as a signal during mating as suggested for other insect species [34]. In addition, an active role as a mating signal, as suggested for other insect species [34], cannot be excluded. In *A. taiwanensis*, the epicuticular waxes may also exert a role in mating. Such a role could be exerted not only as a visual cue, but also as a chemical signal. In fact, a previous study showed that *A. taiwanensis* epicuticular extracts contained a mixture of long-chain paraffines and olefins, long chained 2-ketones, propyl esters of fatty acids and free fatty acids [35], some of them quite uncommon for insects and among Coleoptera. In particular, according to Iovinella et al. [35], a mixture of nonacosene and nonacosadiene is more abundant in males than in females, while n-nonacosane is more abundant in females [35]. Such non-volatile compounds may then exert a role in mating by proximate cue, assessed by the rostral rubbing acts that are performed by the male on the female’s body [36]. In fact, rostral rubbing and antennal tapping are a mating behavior commonly displayed by many Coleoptera species to recognize their potential partners at close range [37]. However, further research is needed to clarify the actual role of the cuticular waxes of *A. taiwanensis*.

## 5. Conclusions

Overall, the protocol developed in this study, based on an artificial diet for the larvae, allowed a complete life cycle of the fig weevil. In particular, no significant differences between the weight and length of wild versus artificially reared adults were observed. This suggests that the artificial diet and rearing protocol are adequate for supporting *A. taiwanensis* development. Being able to rear the fig weevil under laboratory conditions along its complete lifecycle, allowed us to obtain information on biological and behavioral characteristics that may be useful for the development of efficient control programs. However, further studies may be needed to improve the technique to obtain the large numbers of insects necessary for some IPM programs, such as the Sterile Insect Technique, where the large numbers of individuals available is the basis of the strategy.

## Figures and Tables

**Figure 1 insects-14-00223-f001:**
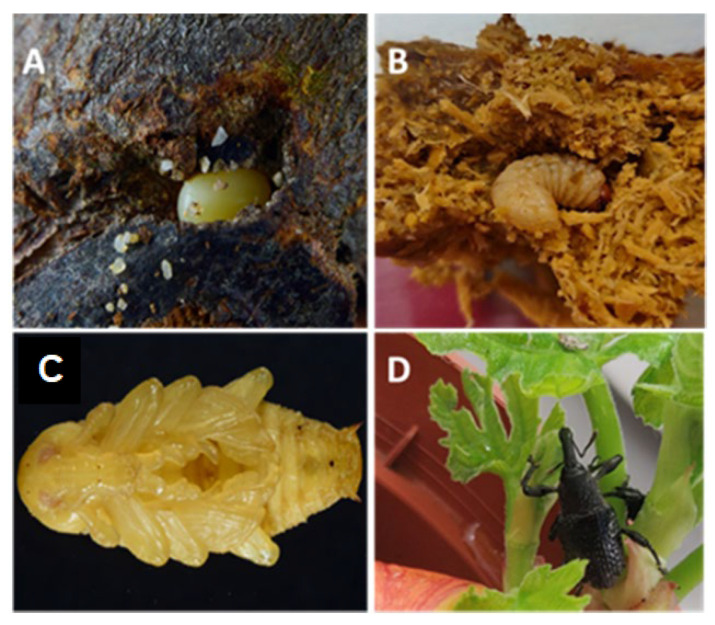
Egg (**A**), larva (**B**), pupa (**C**), and adult (**D**) fig weevil *Aclees taiwanensis*.

**Figure 2 insects-14-00223-f002:**
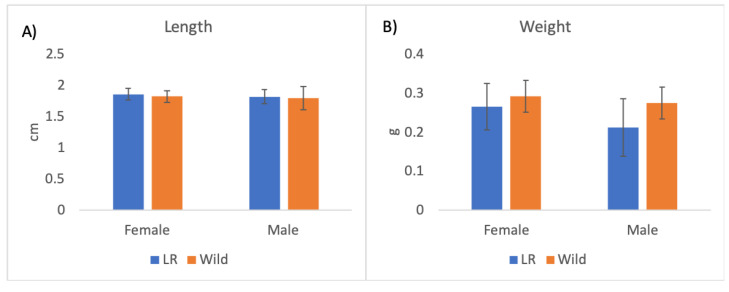
Comparison of body length (**A**), and weight (**B**) between laboratory-reared and wild *Aclees taiwanensis* adults. Histograms represent means. Bars represent standard deviations; n = 30.

**Figure 3 insects-14-00223-f003:**
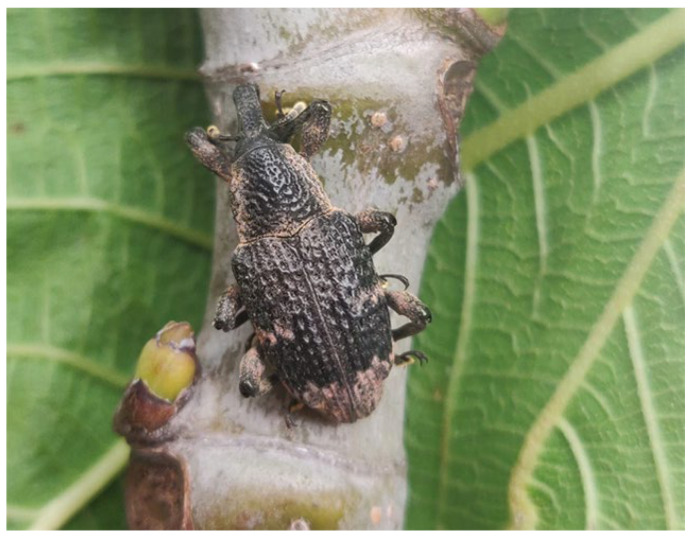
Adult of *Aclees taiwanensis* with patches of the characteristic pinkish beige dust.

**Figure 4 insects-14-00223-f004:**
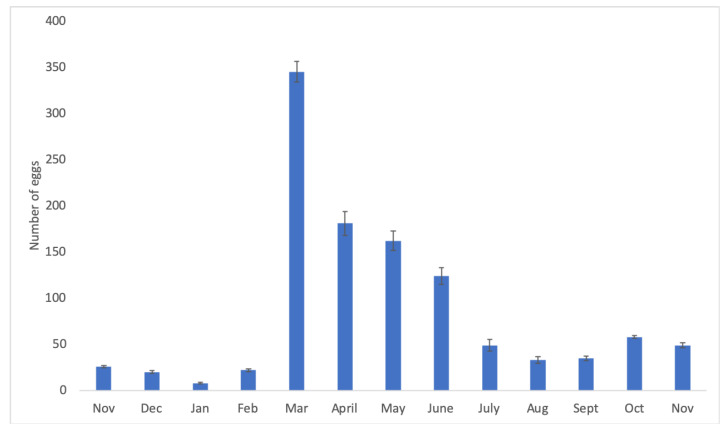
Monthly oviposition (average number of eggs/month/12 females) by *Aclees taiwanensis*. Histograms represent means. Bars represent standard deviations.

**Table 1 insects-14-00223-t001:** *Aclees taiwanensis* reared on an agarized diet: number of eggs laid, and number and percentage of larvae, pupae and adults obtained.

Parameters	Number	% Recorded
Eggs laid	1159	-
Larvae	802	72.12
Pupae	315	39.28
Adults	156	49.52

**Table 2 insects-14-00223-t002:** Main biological traits of laboratory-reared *Aclees taiwanensis*.

Biological Parameter	Mean ± SD
Embryonic duration (days)	8.50 ± 1.92
Total larval duration (days)	41.17 ± 9.56
Pupal duration (days)	12.00 ± 2.25
Number of larval instars	5.00 ± 0.00
Fecundity (eggs/female/year)	96.60 ± 40.30

Data represent means ± standard deviations (SD).

**Table 3 insects-14-00223-t003:** Development time of laboratory *Aclees taiwanensis* pre-imaginal instars reared on an agarized diet.

Instar	Days
L1	3.3 ± 0.80
L2	5.1 ± 2.29
L3	5.6 ± 1.84
L4	7.6 ± 0.99
L5	22.7 ± 2.82

Data represent means ± standard deviations; n = 30.

**Table 4 insects-14-00223-t004:** Morphological parameters of pre-imaginal instars of *Aclees taiwanensis* reared on an agarized diet, and comparison with the length of larvae reared on fig twigs.

Instar	Weight	Length	Length (Reared on Fig Twigs) ^a^	*p*-Value ^b^
L1	-	0.30 ± 0.00	-	
L2	0.01 ± 0.00	0.50 ± 0.06	0.49 ± 0.17	0.790
L3	0.04 ± 0.03	0.72 ± 0.09	0.86 ± 0.41	0.257
L4	0.05 ± 0.01	1.09 ± 0.11	1.06 ± 0.41	0.807
L5	0.40 ± 0.01	1.83 ± 0.13	1.70 ± 0.49	0.301

Data represent means ± standard deviations; n = 30. ^a^ Data from [7]. ^b^ *p*-values of the comparison between the length of larvae reared on an agarized diet and on fig twigs (Student’s *t*-test).

## Data Availability

Data are available on request from the corresponding authors.

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
