# Peer review of "Biological Insights on the Invasive Fig Pest Aclees taiwanensis Kȏno, 1933 (Coleoptera: Curculionidae)"

_insects, 2023, doi:10.3390/insects14030223_

Round 1

Reviewer 1 Report

This is an interesting study that was carried out to determined biological and morphological parameters of an invasive fig weevil. I think this study has field applications to support invasive pest management. A simple summary, abstract, keywords, introduction, objectives, materials & methods, results, and discussion are explained well. Discussion is too wordy and need precision. No conclusion section is provided. I suggest authors to develop a paragraph on conclusion. 

Following are other comments and suggestions: 

Title: use the Caps throughout the title

Line 5, 7, and 9: delete DAFE

Line 16: delete forms and change larval to larva

Line 17: delete and insert a period after the plant. Start a new sentence with To date

Line 18: delete that and rephrase this sentence. 

Line 23: delete extra space

Line 24: delete ‘and scenic beauty’ 

Line 30: delete very scarce

Line 44: bring also before reported

Line 57: delete administration and insert delivery

Line 72 and 74: delete pot and insert cage

Line 123: delete Female

Line 137: insert period

Table 1: Headings…First column: Parameters, Second column Number, Third column % Recorded

Line 158 complete this sentence and adjust in the narrative

Table 4: First column, delete Female; Second Column insert Mean±SD

Figure 1: Correct C

Line 173: delete to and insert into; delete and. Start new sentence with About

Line 177: delete consequent and insert they died premature

Line 180: delete cuticula and insert cuticle

Line 181: change whitish to white in color

Figure 2: Correct the spellings; it should read Body length, and Body weight. Also, Y-Axis correct the length in cm and weight in g

Line 206: Delete Female

Figure 4: Y-Axis, change n. eggs to Number of eggs (Mean±SD)

Line 221: delete ‘the species’

Line 222: after described insert period; Start new sentence with Also

Line 224: delete although successful

Line 227: check in Table 1 if 61% is correct

Line 237: remove extra space before comma

Line 239: rephrase this sentence

Line 278: delete thanks to

Line 279: before weevil insert a period and start new sentence with, We

Author Response

This is an interesting study that was carried out to determined biological and morphological parameters of an invasive fig weevil. I think this study has field applications to support invasive pest management. A simple summary, abstract, keywords, introduction, objectives, materials & methods, results, and discussion are explained well. Discussion is too wordy and need precision. No conclusion section is provided. I suggest authors to develop a paragraph on conclusion.

R: We are very grateful to the Reviewer for the expert revision of the manuscript. We re-worded parts of the Discussion and added a Conclusion section. Following his suggestions and comments we, point-by-point, corrected and modified the manuscript which is now, in our opinion, much improved.

Following are other comments and suggestions:

Title: use the Caps throughout the title

R: Done

Line 5, 7, and 9: delete DAFE

R: Done

Line 16: delete forms and change larval to larva

R: Done

Line 17: delete and insert a period after the plant. Start a new sentence with To date

R: Done

Line 18: delete that and rephrase this sentence.

R: Done

Line 23: delete extra space

R: Done

Line 24: delete ‘and scenic beauty’

R: Done

Line 30: delete very scarce

R: Done

Line 44: bring also before reported

R: done

Line 57: delete administration and insert delivery

R: Done

Line 72 and 74: delete pot and insert cage

R: Actually, for the experiment, we inserted a pot filled with soil in each cage in order to allow the insect to stay sheltered underground. This was also because, in nature, during the day the adults are often hidden at the base of the trunk where the roots emerge from the ground.

Line 123: delete Female

R: Done

Line 137: insert period.

R: We completed the information about the software used.

Table 1: Headings…First column: Parameters, Second column Number, Third column % Recorded

R: Done

Line 158 complete this sentence and adjust in the narrative

R: Done

Table 4: First column, delete Female; Second Column insert Mean±SD

R: Done

Figure 1: Correct C

R: Done

Line 173: delete to and insert into; delete and. Start new sentence with About

R: Done

Line 177: delete consequent and insert they died premature

R: Done

Line 180: delete cuticula and insert cuticle

R: Done

Line 181: change whitish to white in color

R: Done

Figure 2: Correct the spellings; it should read Body length, and Body weight. Also, Y-Axis correct the length in cm and weight in g

R: we corrected Figure 2

Line 206: Delete Female

R: Done

Figure 4: Y-Axis, change n. eggs to Number of eggs (Mean±SD)

R: Done

Line 221: delete ‘the species’

R: Done

Line 222: after described insert period; Start new sentence with Also

R: Done

Line 224: delete although successful

R: Done

Line 227: check in Table 1 if 61% is correct

R: We corrected the value

Line 237: remove extra space before comma

R: Done

Line 239: rephrase this sentence

R: Done

Line 278: delete thanks to

R: Done

Line 279: before weevil insert a period and start new sentence with, We

R: Done

Reviewer 2 Report

A very good study, providing new, interesting and valuable information on rearing weevils in the lab. 

I found the methods and results convincing.

My only concern is about the English language. In many places, the text appears to have been translated from another language, using a web-based translation service (e.g., Google Translate), and some of the sentences are not constructed properly. This affects the readability. I would suggest getting the paper rewritten in correct English. Perhaps someone who is well-versed in scientific writing in English may be able to help. Please have this done, because this study should certainly be published. Best of luck!

Author Response

A very good study, providing new, interesting and valuable information on rearing weevils in the lab. 

I found the methods and results convincing.

My only concern is about the English language. In many places, the text appears to have been translated from another language, using a web-based translation service (e.g., Google Translate), and some of the sentences are not constructed properly. This affects the readability. I would suggest getting the paper rewritten in correct English. Perhaps someone who is well-versed in scientific writing in English may be able to help. Please have this done, because this study should certainly be published. Best of luck!

R: We thank very much the Reviewer for the comments and suggestions. After this round of revision, we will ask the Journal for a language editing service to check and correct the English of the manuscript.

Author Response

Manuscript ID: insects-2207247 Journal: Insects

Title:

(Coleoptera: Curculionidae).

The manuscript illustrates the possibility of rearing the beetle Aclees taiwanensis, an invasive pest of the fig tree, the rearing conditions and their possible influence on the insect's main growth parameters. See its ability to damage a very common and useful plant of the Mediterranean area, it is important to know details of its biology and its behaviour, so that invasions can be consciously controlled. The aim of this paper is clearly to fill an information gap on this species.

Strong points: Development of a laboratory protocol suitable for the Aclees taiwanensis rearing that allows the main fitness parameters of the species to be assessed.

Knowledge of the main biological characteristics of the insect is useful for the development of strategies to control the invasion of a plant that is very useful to humans.

Weak points: The introduction is a little too brief and does not clarify the many information gaps which, if emphasised, would give greater emphasis to the work done. That there are not many articles in the literature on this subject is also evident from the bibliography: more than 50% of the articles cited are older than 10 years. Suggestions: Explain in the introduction what little information is available, describe at least briefly the insect's distribution area, highlight the information gaps that exist, and emphasise the importance of your own contribution to filling them.

R: We thank the Reviewer very much for the valuable and very useful suggestions. We have re-written the introduction accordingly

General Notes:

By centring the figures on the page, it is possible to place the captions on whole lines and not interrupted in the middle (Fig. 3 - R198-199, Fig. 4 - R213-214); R246: Why not report these data already in the results? You can put them in a table with those obtained by you, so that you can more easily compare them, even visually.

R: as suggested we added in Table 2 the data about larvae length reared in fig twigs.

A simple statistical discussion could then be added to support the statement in R246-247 ("suggesting that the artificial diet did not influence this aspect of larval development").

R: as suggested we performed a statistical comparison between artificial diet and fig twigs reared larvae and we added the results in table 2 and in the results section (lines 169-171).

Specific Comments:

R73 Replace 'A fig a woody sprig' with 'A fig's woody spring'.

R: Changed. Also as suggested by REVIEWER 2.

R158 This sentence is completely detached from the text, probably it s a typo, because line 154, of which 158 should be the continuation after the table, ends with a full stop. Remove it or write the entire sentence.

R: We corrected the sentence. Subject (“Duration”) was missing. We apologize for that.

R158-167 Adjust the font size used, otherwise it is not clear what is referred to in the tables and what is part of the main text.

R: Done.

R193-194 Insert a line between the caption of figure 2 and the main text.

R: Done

R209 A standard deviation of 40.30 out of a mean of 96.60??? I think there was some problem with the commas and 0s because in the graph the error does not seem so consistent... such a standard deviation means that the number of eggs can vary from 56 to 137! Review the reported standard deviation or the calculations that determined it.

R: We thank the Reviewer for having spotted the mistake. We have now corrected the value (line 232-233).

Reviewer 4 Report

Remarks to the Editor and Authors:

I would like to thank the Editor for the opportunity to review this manuscript. This was an interesting and very relevant study with regards to a recently introduced insect pest species, Aclees taiwanensis, that is threatening fig nurseries, orchards, and wild plants of southern Europe. As pointed out by the authors, the biology and life-history characteristics of this weevil is not well-known, and is very relevant for developing appropriate control methods. This research is significant and appropriate in light of the enhanced rates of introduction and spread of new invasive pests due to trade and movement, as well as climate change.

Goals in this study were: i) to develop a rearing technique for A. taiwanensis based on the use of an artificial diet; and ii) to study the main features of the weevil’s biology.

The Materials & Methods section was clear and well explained.

Most of the Results section and Tables and Figures were clear and well explained. I have only a few minor points that I believe need to be addressed (See Comments & Suggested Changes below and specifically my comment on Figure 4).

The Discussion section was for the most part well supported by the study results. I do have some remarks and suggestions that can be found below in my Comments & Suggested Changes that I believe will strengthen the manuscript.

Comments & Suggested Changes:

Simple Summary:

Line 16: Replace “larval forms” with “larval stages”.

Introduction:

Line 51: Replace “are” with “it is”.

Materials and Methods:

Line 67-68: What time of the year were the wild specimens (3 couples), that were used to start the colony with, collected from the field? It may be relevant to researchers who would like to conduct similar studies.

Line 74 and 106: Fellow researchers are always interested in the specifics of rearing methods, could you therefore please provide a little more detail on how water was provided? (For example, as moistened cotton wool in a vial, or as a light spray applied daily?)

Line 73: Suggesting that “A fig a woody sprig” should be reworded as “A woody twig of the host plant” or “A woody fig twig”. It would be useful to also give a measurement of the approximate diameter of the woody twig.

Line 74: Replace “insert” with “inserted”.

Line 86: Replace “let cool up to 50 °C” with “allowed to cool down to 50 °C”.

Line 96: Suggest replacing “newborn” with “newly emerged”.

Line 109: Suggest replacing “pupal one” with “pupal stage”.

Line 132: Suggest replacing “newborn” with “newly emerged”.

Results:

Lines 141-142: Suggest rewording the sentence to: “Of the 800 larvae fed on the artificial diet, about 39.3% pupated, and from these pupae 49.5% adults developed (Table 1)”.

Lines 145-146: Suggest rewording the sentence to: “Besides the other ingredients, we modulated the preservatives to inhibit mould and bacterial growth, while allowing the survival of the larvae”.

Line 151: Suggest rewording the sentence to: “When reared on the artificial diet formulated in this experiment,…..”.

Line 158: Insert “Duration” at the beginning of the sentence.

Line 184: Suggest rewording to: “Among the adults that emerged from the reared pupae, females represented...”.

Lines 186-190: Suggest restructuring these sentences. The finding that weight and length of reared adults did not significantly differ from that of wild specimens is important and indicates that the artificial diet is suitable. I suggest it should therefore be a separate, stand-alone sentence: “Though males were significantly lighter than females (t-test, t = 3.101, df = 58, p = 0.003), no significant differences in body length were found between the sexes (t-test, t = 1.817, df = 60, p = 0.074). No significant differences in the weight and length of females and males were found between wild specimens and adults reared on the agarised diet (AD) (t = 1.691, df = 60, p = 0.097) (Figure 2).

Figure 2: Please note spelling mistake in Graph title: “Weigh” should be changed to “Weight”

Line 208: Females laid their first eggs after 71.00 ± 10.06 days from emergence – Was this from the date of adult or larval emergence? May be a good idea to specify.

Line 209: Suggest replacing “spawning” with “egg production”, and “an average eggs production of…” with “an average oviposition rate of….”.

Lines 208-211 and Figure 4: I am not sure if I understand correctly; it is reported that the average number of eggs produced by a female is 96.6 per female/year (Table 4). Figure 4 though, indicates about 350 eggs/month/female just in the month of March. Is Fig. 4’s caption, stating “Monthly oviposition (average number of eggs/month/female) by A. taiwanensis couples” correct? Or should it be the total number of eggs laid by all couples (thus for all 12 females in total) used in the study? Sorry, it is just not clear to me.

Discussion:

Line 250: Replace “temperature” with “temperatures”.

Lines 263-264: Did you study specific morphological characteristics in the pupae? It would be best to add a short description of your methods and findings in the Materials & Methods and Results sections to support this observation mentioned in your discussion.

Lines 272-277: In this paragraph the authors state “In our study, biological parameters such as egg hatching rate (Table 1), larval and pupal duration (Table 4) are similar to those observed in other Curculionidae [36,37,30] under about the same environmental conditions”. Ref 36 (Beavers 1982) reported that for Diaprepes abbreviates, the mean developmental periods for larvae and pupae, respectively, were 377 days and 15.2 days for females and 378 days and 15.4 days for males. Ref 37 (Kaakeh 2005) reported that the developmental times for larvae of Rynchophorus ferrugineus ranged from 70.8 to 102.2 days, while the development time of pupae ranged from 16.1 to 22.2 days. In the current study the authors report that larval development lasts 41.17 days and pupal development 12.0 days. This is not similar to the results of the cited references and I suggest that this paragraph be removed. An important finding of this study was that the weights and lengths of wild vs reared specimens did not differ significantly, suggesting that the diet and rearing was adequate. I suggest incorporating this fact into the final paragraph of the paper: “This study found no significant differences between the weight and length of wild versus artificially reared adults, which suggest that the artificial diet and rearing protocol developed, are adequate for supporting A. taiwanensis.” Being able to rear the complete lifecycle of the fig weevil under laboratory conditions, allowed us to obtain information on biological and behavioural characteristics that may be useful for the development of efficient control programs. However, further studies may be needed to improve the technique to obtain the large numbers of insects that may be needed for some IPM programs such as Sterile Insect Technique, where the rearing of large numbers of individuals is the basis of the strategy.”

References:

Line 363: Reference 35 should be changed to: “Nielsen, E. T., & Evans, D. G. (1960). Duration of the pupal stage of Aedes taeniorhynchus with a discussion of the velocity of development as a function of temperature. Oikos11(2), 200-222.”

Thank you

Author Response

Remarks to the Editor and Authors:

I would like to thank the Editor for the opportunity to review this manuscript. This was an interesting and very relevant study with regards to a recently introduced insect pest species, Aclees taiwanensis, that is threatening fig nurseries, orchards, and wild plants of southern Europe. As pointed out by the authors, the biology and life-history characteristics of this weevil is not well-known, and is very relevant for developing appropriate control methods. This research is significant and appropriate in light of the enhanced rates of introduction and spread of new invasive pests due to trade and movement, as well as climate change.

Goals in this study were: i) to develop a rearing technique for A. taiwanensis based on the use of an artificial diet; and ii) to study the main features of the weevil’s biology.

The Materials & Methods section was clear and well explained.

Most of the Results section and Tables and Figures were clear and well explained. I have only a few minor points that I believe need to be addressed (See Comments & Suggested Changes below and specifically my comment on Figure 4).

The Discussion section was for the most part well supported by the study results. I do have some remarks and suggestions that can be found below in my Comments & Suggested Changes that I believe will strengthen the manuscript.

R: We thank the Reviewer very much for the valuable comments and suggestions which have helped us to revise and substantially improve the manuscript.

Comments & Suggested Changes:

Simple Summary:

Line 16: Replace “larval forms” with “larval stages”.

R: Done. We replaced “larval forms” with “larvae” as suggested also by Reviewer 1

Introduction:

Line 51: Replace “are” with “it is”.

R: Done.

Materials and Methods:

Line 67-68: What time of the year were the wild specimens (3 couples), that were used to start the colony with, collected from the field? It may be relevant to researchers who would like to conduct similar studies.

R: Done, the specimens have been collected in May 2019

Line 74 and 106: Fellow researchers are always interested in the specifics of rearing methods, could you therefore please provide a little more detail on how water was provided? (For example, as moistened cotton wool in a vial, or as a light spray applied daily?)

R: Done, water was supplied by means of a roll of gauze which dipped into a falcon test tube with a pierced stopper.

Line 73: Suggesting that “A fig a woody sprig” should be reworded as “A woody twig of the host plant” or “A woody fig twig”. It would be useful to also give a measurement of the approximate diameter of the woody twig.

R: Done.

Line 74: Replace “insert” with “inserted”.

R: Done.

Line 86: Replace “let cool up to 50 °C” with “allowed to cool down to 50 °C”.

R: Done.

Line 96: Suggest replacing “newborn” with “newly emerged”.

R: Done.

Line 109: Suggest replacing “pupal one” with “pupal stage”.        

R: Done.

Line 132: Suggest replacing “newborn” with “newly emerged”.

R: Done.

Results:

Lines 141-142: Suggest rewording the sentence to: “Of the 800 larvae fed on the artificial diet, about 39.3% pupated, and from these pupae 49.5% adults developed (Table 1)”.

R: Done.

Lines 145-146: Suggest rewording the sentence to: “Besides the other ingredients, we modulated the preservatives to inhibit mould and bacterial growth, while allowing the survival of the larvae”.

R: Done.

Line 151: Suggest rewording the sentence to: “When reared on the artificial diet formulated in this experiment,…..”.

R: Done.

Line 158: Insert “Duration” at the beginning of the sentence.

R: Done.

Line 184: Suggest rewording to: “Among the adults that emerged from the reared pupae, females represented...”.

R: Done

Lines 186-190: Suggest restructuring these sentences. The finding that weight and length of reared adults did not significantly differ from that of wild specimens is important and indicates that the artificial diet is suitable. I suggest it should therefore be a separate, stand-alone sentence: “Though males were significantly lighter than females (t-test, = 3.101, df = 58, = 0.003), no significant differences in body length were found between the sexes (t-test, = 1.817, df = 60, p = 0.074). No significant differences in the weight and length of females and males were found between wild specimens and adults reared on the agarised diet (AD) (= 1.691, df = 60, p = 0.097) (Figure 2).

R: Done. We changed the sentence as suggested

Figure 2: Please note spelling mistake in Graph title: “Weigh” should be changed to “Weight”

R: We have corrected the graph title

Line 208: Females laid their first eggs after 71.00 ± 10.06 days from emergence – Was this from the date of adult or larval emergence? May be a good idea to specify.

R: Done, we added “71.00 ± 10.06 days after emergence from the pupal stage”

Line 209: Suggest replacing “spawning” with “egg production”, and “an average eggs production of…” with “an average oviposition rate of….”.

R: Done

Lines 208-211 and Figure 4: I am not sure if I understand correctly; it is reported that the average number of eggs produced by a female is 96.6 per female/year (Table 4). Figure 4 though, indicates about 350 eggs/month/female just in the month of March. Is Fig. 4’s caption, stating “Monthly oviposition (average number of eggs/month/female) by A. taiwanensis couples” correct? Or should it be the total number of eggs laid by all couples (thus for all 12 females in total) used in the study? Sorry, it is just not clear to me.

R: We apologize. There was a mistake in the caption of Figure 4. We have now corrected it

Discussion:

Line 250: Replace “temperature” with “temperatures”.

R: Done

Lines 263-264: Did you study specific morphological characteristics in the pupae? It would be best to add a short description of your methods and findings in the Materials & Methods and Results sections to support this observation mentioned in your discussion.

R: The pupal stage is very delicate and does not allow for manipulation. Any attempt to measure length, weight or to observe in deep the pupae had a very negative impact for their survival. The only observations allowed concerned a delicate general observation without manipulation of the external morphology from which it was possible to establish no sexual dimorphism. We did not report this consideration in the text unless you have a different opinion.

Lines 272-277: In this paragraph the authors state “In our study, biological parameters such as egg hatching rate (Table 1), larval and pupal duration (Table 4) are similar to those observed in other Curculionidae [36,37,30] under about the same environmental conditions”. Ref 36 (Beavers 1982) reported that for Diaprepes abbreviates, the mean developmental periods for larvae and pupae, respectively, were 377 days and 15.2 days for females and 378 days and 15.4 days for males. Ref 37 (Kaakeh 2005) reported that the developmental times for larvae of Rynchophorus ferrugineus ranged from 70.8 to 102.2 days, while the development time of pupae ranged from 16.1 to 22.2 days. In the current study the authors report that larval development lasts 41.17 days and pupal development 12.0 days. This is not similar to the results of the cited references and I suggest that this paragraph be removed.

R: Done

An important finding of this study was that the weights and lengths of wild vs reared specimens did not differ significantly, suggesting that the diet and rearing was adequate. I suggest incorporating this fact into the final paragraph of the paper: “This study found no significant differences between the weight and length of wild versus artificially reared adults, which suggest that the artificial diet and rearing protocol developed, are adequate for supporting A. taiwanensis.” Being able to rear the complete lifecycle of the fig weevil under laboratory conditions, allowed us to obtain information on biological and behavioural characteristics that may be useful for the development of efficient control programs. However, further studies may be needed to improve the technique to obtain the large numbers of insects that may be needed for some IPM programs such as Sterile Insect Technique, where the rearing of large numbers of individuals is the basis of the strategy.”

R: Done

References:

Line 363: Reference 35 should be changed to: “Nielsen, E. T., & Evans, D. G. (1960). Duration of the pupal stage of Aedes taeniorhynchus with a discussion of the velocity of development as a function of temperature. Oikos11(2), 200-222.”

R: The Ref. was deleted according to the Reviewer’s suggestions

Thank you

Round 2

Reviewer 1 Report

The manuscript has been improved after the revision.